# MAP Propagation Algorithm: Faster Learning with a Team of Reinforcement Learning Agents

**Stephen Chung**
Department of Computer Science
University of Massachusetts Amherst
Amherst, MA 01003
minghaychung@umass.edu

## Abstract

Nearly all state-of-the-art deep learning algorithms rely on error backpropagation, which is generally regarded as biologically implausible. An alternative way of training an artificial neural network is through treating each unit in the network as a reinforcement learning agent, and thus the network is considered as a team of agents. As such, all units can be trained by REINFORCE, a local learning rule modulated by a global signal that is more consistent with biologically observed forms of synaptic plasticity. Although this learning rule follows the gradient of return in expectation, it suffers from high variance and thus the low speed of learning, rendering it impractical to train deep networks. We therefore propose a novel algorithm called MAP propagation to reduce this variance significantly while retaining the local property of the learning rule. Experiments demonstrated that MAP propagation could solve common reinforcement learning tasks at a similar speed to backpropagation when applied to an actor-critic network. Our work thus allows for the broader application of teams of agents in deep reinforcement learning.

## 1 Introduction

*Error backpropagation algorithm* (backprop) [1] efficiently computes the gradient of an objective function with respect to parameters, by iterating backward from the last layer of a multi-layer artificial neural network (ANN). However, backprop is generally regarded as being biologically implausible [2, 3, 4, 5, 6, 7]. First, the learning rule given by backprop is non-local, as it relies on information other than input and output of a neuron-like unit computed in the feedforward phase; while biologically-observed synaptic plasticity depends mostly on local information (e.g. spike-timing-dependent plasticity (STDP) [8]) and possibly some global signals (e.g. reward-modulated spike-timing-dependent plasticity (R-STDP) [8, 9, 10]). Second, backprop requires precise coordination between feedforward and feedback connections, because the feedforward value has to be retained until error signals arrive; while it is unclear how a biological system can coordinate an entire network to alternate between feedforward and feedback phases precisely. Third, backprop requires synaptic symmetry in the forward and backward paths, rendering it biologically implausible. Nonetheless, recent work has demonstrated that this symmetry may not be necessary for backprop due to the 'feedback alignment' phenomenon [11, 12, 13].

Alternatively, REINFORCE [14] could be applied to all units in the network to train an ANN as a more biologically plausible way of learning. It is shown that the learning rule gives an unbiased estimate of the gradient of return [14]. Another interpretation of this relates to viewing each unit as a reinforcement learning (RL) agent, with each agent trying to maximize the global reward. Such a *team of agents* is also known as a *coagent network* [15]. However, coagent networks can only solve

simple tasks due to the high variance associated with the learning rule and thus the low speed of learning. The high variance stems from the lack of structural credit assignment, i.e. a single scalar reward is used to evaluate the action of all agents in the network.

To address this high variance associated with REINFORCE, we propose a novel algorithm that significantly reduces the variance while retaining the local property of the learning rule. We call this newly proposed algorithm *maximum a posteriori (MAP) propagation*. Essentially, MAP propagation replaces the hidden units' values with their MAP estimates conditioned on the action chosen, or equivalently, minimizes the energy function of the network, before applying REINFORCE. We prove that for a network with normally distributed hidden units, by minimizing the energy function of the network, the parameter update given by REINFORCE and backprop (with the reparametrization trick) becomes the same, thus establishing a connection between REINFORCE and backprop. Our experiments show that a team of agents trained with MAP propagation can learn much faster than REINFORCE, such that the team of agents can solve common RL tasks at a similar (or higher) speed compared to an ANN trained by backprop, as well as exhibiting sophisticated exploration that differs from an ANN trained by backprop.

The novel MAP propagation algorithm represents a new class of algorithm to train an ANN that is more biologically plausible than backprop and maintains a comparable learning speed to backprop at the same time. Our work also opens the prospect of the broader application of teams of agents, called coagent networks [15], in deep RL.

## 2  Background and Notation

We consider a Markov Decision Process (MDP) defined by a tuple $(\mathcal{S}, \mathcal{A}, P, R, \gamma, d_0)$, where $\mathcal{S}$ is a finite set of states of an agent's environment (although this work can be extended to the infinite state case), $\mathcal{A}$ is a finite set of actions, $P : \mathcal{S} \times \mathcal{A} \times \mathcal{S} \to [0, 1]$ is a transition function giving the dynamics of the environment, $R : \mathcal{S} \times \mathcal{A} \to \mathbb{R}$ is a reward function, $\gamma \in [0, 1]$ is a discount factor, and $d_0 : \mathcal{S} \to [0, 1]$ is an initial state distribution. Denoting the state, action, and reward signal at time $t$ by $S_t$, $A_t$, and $R_t$ respectively, $P(s, a, s') = \Pr(S_{t+1} = s'|S_t = s, A_t = a)$, $R(s, a) = \mathbb{E}[R_t|S_t = s, A_t = a]$, and $d_0(s) = \Pr(S_0 = s)$, where $P$ and $d_0$ are valid probability mass functions. An episode is a sequence of states, actions, and rewards, starting from $t = 0$ and continuing until reaching the terminal state. For any learning methods, we can measure its performance as it improves with experience over multiple episodes, which makes up a run.

Letting $G_t = \sum_{k=t}^{\infty} \gamma^{k-t} R_k$ denote the infinite-horizon discounted return accrued after acting at time $t$, we are interested in finding, or approximating, a *policy* $\pi : \mathcal{S} \times \mathcal{A} \to [0, 1]$ such that for any time $t \geq 0$, selecting actions according to $\pi(s, a) = \Pr(A_t = a|S_t = s)$ maximizes the expected return $\mathbb{E}[G_t|\pi]$. The value function for policy $\pi$ is $V^\pi$ where for all $s \in \mathcal{S}$, $V^\pi(s) = \mathbb{E}[G_t|S_t = s, \pi]$, which can be shown to be independent of $t$ for the infinite-horizon case we are considering.

Here we restrict attention to policies computed by multi-layer networks consisting of $L$ layers of stochastic units. Let $H_t^l \in \mathbb{R}^{n(l)}$ denote the activation values of the units in layer $l$ at time $t$ and $n(l)$ denote the number of units in layer $l$. For any $t \geq 0$, we also let $H_t^0 = S_t$, $H_t^L = A_t$, and $H_t = \{H_t^1, H_t^2, ..., H_t^{L-1}\}$. We call any elements in $H_t$ a hidden layer and $H_t^L$ the output layer. The distribution of $H_t^l$ conditional on $H_t^{l-1}$ is given by $\pi_l : \mathbb{R}^{n(l-1)} \times \mathbb{R}^{n(l)} \to [0, 1]$, such that for any $t \geq 0$, $\pi_l(h^{l-1}, h^l; W^l) = \Pr(H_t^l = h^l|H_t^{l-1} = h^{l-1}; W^l)$, where $W^l$ is the parameter of layer $l$. We also denote all parameters of the network as $W = \{W^1, W^2, ..., W^L\}$. To sample an action $A_t$ from the network, we iteratively sample $H_t^l \sim \pi_l(H_t^{l-1}, \cdot; W^l)$ from $l = 1$ to $L$.

We say a layer $l$ is normally distributed if $\pi_l(H_t^{l-1}, \cdot; W^l) = N(g^l(H_t^{l-1}; W^l), \sigma_l^2)$, the normal distribution with mean $g^l(H_t^{l-1}; W^l)$, where $g^l : \mathbb{R}^{n(l-1)} \to \mathbb{R}^{n(l)}$ is a differentiable function, and a fixed standard deviation $\sigma_l$. A common choice of $g$ is a linear transformation followed by an activation function; that is, $g(H_t^{l-1}; W^l) = f(W^l H_t^{l-1})$ where $f$ is a non-linear activation function such as softplus or rectified linear unit (ReLU) and $W^l \in \mathbb{R}^{n(l) \times n(l-1)}$. We also define the *energy function* $E : \mathbb{R}^{n(1)} \times \mathbb{R}^{n(2)} \times ... \times \mathbb{R}^{n(L-1)} \to [0, \infty)$ as $E(h; s, a) = -\log \Pr(H_t = h|S_t = s, A_t = a)$, which can be shown to be independent of $t$.

The case we consider here is one in which all the units of the network implement an RL algorithm and share the same reward signal. These networks can therefore be considered as *teams of agents* (agent here refers to an *RL agent* [16]), which have also been called *coagent networks* [15].

We denote $\nabla_x f$ as the gradient of $f$ w.r.t. $x$, $A^T$ as the transpose of matrix $A$, and $\nabla_A f(\Pr(A))$ as the shorthand for $\nabla_a f(\Pr(A = a))$. For a random variable $X$ with a distribution that depends on parameter $W$ and a random variable $Y$, we call $h(Z; W, Y)$ the *re-parameterization of $X$ by $Z$ conditioned on $Y$* if $h(Z; W, Y)$ and $X$ have the same conditional distribution; that is, $\Pr(h(Z; W, Y) = x | Y = y) = \Pr(X = x | Y = y; W)$ for all values of $x$, $y$ and $W$, where $Z$ is an independent random variable with a distribution that does not depend on parameter $W$ and $h$ is an invertible and differentiable function. In case $X$ has a multi-layer structure, we denote $h^l(Z; W, Y)$ as the $l^{\text{th}}$ layer in $h(Z; W, Y)$ and $h^{-1}$ as the inverse of $h$. In general, we use the superscript $l$ to denote the $l^{\text{th}}$ layer in a variable if the variable has a multi-layer structure. Also, for all distributions discussed in this paper, the probability mass function is replaced by probability density function if the random variable is continuous.

## 3 Algorithm

### 3.1 MAP Propagation

MAP propagation is based on REINFORCE applied to each hidden unit with the same global reinforcement signal. To reduce the variance associated with the learning rule, we note that this variance can be reduced by using the expected parameter update conditioned on the state and the selected action instead. However, this expected parameter update is generally intractable to compute analytically. Therefore, we propose to use the *maximum a posteriori* (MAP) estimate to approximate the expected parameter update. This makes the resulting learning rule biased but reduces the variance significantly. The details of MAP propagation are as below.

The gradient of return with respect to $W^l$ (where $l \in \{1, 2, ..., L\}$ in all discussion below unless stated otherwise) can be estimated by REINFORCE, also known as likelihood ratio estimators:

$$\nabla_{W^l} \mathbb{E}[G_t] = \sum_{k=t}^{\infty} \gamma^{(k-t)} \mathbb{E}[G_k \nabla_{W^l} \log \Pr(A_k | S_k)]. \tag{1}$$

We can show that the terms in the summation of (1) also equal $\mathbb{E}[G_k \nabla_{W^l} \log \pi_l(H_k^{l-1}, H_k^l; W^l)]$, which is the REINFORCE learning rule applied to each hidden unit with the same global reinforcement signal $G_k$:

**Theorem 1.** *Let the policy be a multi-layer network of stochastic units as defined in Section 2. For any $t \geq 0$ and $l \in \{1, 2, ..., L\}$,*

$$\mathbb{E}[G_t \nabla_{W^l} \log Pr(A_t | S_t; W)] = \mathbb{E}[G_t \nabla_{W^l} \log \pi_l(H_t^{l-1}, H_t^l; W^l)]. \tag{2}$$

The proof is in Appendix B.1. Note that this theorem is also proved by Williams [14]. This shows that we can apply REINFORCE to each unit of the network, and the learning rule still gives an unbiased estimate of the gradient of the return. Therefore, denoting $\alpha$ as the step size, we can update parameters by the following stochastic gradient ascent rule:

$$W^l \leftarrow W^l + \alpha G_t \nabla_{W^l} \log \pi_l(H_t^{l-1}, H_t^l; W^l). \tag{3}$$

However, this learning rule suffers from high variance since a single reward, which results from the exploration of all units, is used to evaluate actions of all units, making the learning rule scale poorly with the number of units in the network. To reduce the variance, we notice that we can replace $\nabla_{W^l} \log \pi_l(H_t^{l-1}, H_t^l; W^l)$ in learning rule (3) by $\mathbb{E}[\nabla_{W^l} \log \pi_l(H_t^{l-1}, H_t^l; W^l) | S_t, A_t]$, noting that (see Appendix B.1 for the details; at the R.H.S., the outer expectation is taken over $S_t$, $A_t$ and $G_t$, while the inner expectaion is taken over $H_t^{l-1}$ and $H_t^l$):

$$\mathbb{E}[G_t \nabla_{W^l} \log \pi_l(H_t^{l-1}, H_t^l; W^l)] = \mathbb{E}[G_t \mathbb{E}[\nabla_{W^l} \log \pi_l(H_t^{l-1}, H_t^l; W^l) | S_t, A_t]]. \tag{4}$$

This can reduce variance since the variance associated with the stochastic noise of hidden units is removed in the learning rule (see Appendix B.6 for the proof). The learning rule now becomes:

$$W^l \leftarrow W^l + \alpha G_t \mathbb{E}[\nabla_{W^l} \log \pi_l(H_t^{l-1}, H_t^l; W^l) | S_t, A_t]. \tag{5}$$

Since (3) is following gradient of return in expectation, and the expected update value of (3) and (5) is the same, we conclude that (5) is also a valid learning rule as it follows the gradient of return in expectation. However, we note that $\mathbb{E}[\nabla_{W^l} \log \pi_l(H_t^{l-1}, H_t^l; W^l)|S_t, A_t]$ in (5) is generally intractable to compute analytically. Instead, we propose to use the MAP estimate to approximate this term:[1]

$$\mathbb{E}[\nabla_{W^l} \log \pi_l(H_t^{l-1}, H_t^l; W^l)|S_t, A_t] \approx \nabla_{W^l} \log \pi_l(\hat{H}_t^{l-1}, \hat{H}_t^l; W^l), \qquad (6)$$

where $\hat{H}_t = \mathrm{argmax}_{h_t} \Pr(H_t = h_t|S_t, A_t)$. There are many methods to approximate $\hat{H}_t$, such as hill-climbing methods. In case of hidden units being continuous, we can approximate $\hat{H}_t$ by running gradient ascent on $\log \Pr(H_t|S_t, A_t)$ as a function of $H_t$ for fixed $S_t$ and $A_t$, such that $H_t$ approaches $\hat{H}_t$. $H_t$ can be initialized as the value sampled from the network when sampling action $A_t$. To be concrete, before applying learning rule (3), we first run gradient ascent on $H_t$ for $N$ steps:

$$H_t \leftarrow H_t + \alpha \nabla_{H_t} \log \Pr(H_t|S_t, A_t). \qquad (7)$$

For $l \in \{1, 2, ..., L-1\}$, this is equivalent to (see Appendix B.4 for the details):

$$H_t^l \leftarrow H_t^l + \alpha(\nabla_{H_t^l} \log \pi_{l+1}(H_t^l, H_t^{l+1}; W^{l+1}) + \nabla_{H_t^l} \log \pi_l(H_t^{l-1}, H_t^l; W^l)). \qquad (8)$$

This update rule maximizes the probability of hidden units' value given the state and the action with respect to hidden units' value. Based on the definition of *energy functions* in Section 2, the update rule of hidden units can also be seen as minimizing the energy function $E(H_t; S_t, A_t)$ [17].

After updating $H_t$ for $N$ steps by (8), we obtain an estimate of $\hat{H}_t$, denoted as $\tilde{H}_t$, and apply the following learning rule to learn the parameters of network:

$$W^l \leftarrow W^l + \alpha G_t \nabla_{W^l} \log \pi_l(\tilde{H}_t^{l-1}, \tilde{H}_t^l; W^l). \qquad (9)$$

We call the algorithm that applies REINFORCE after replacing the value of hidden units by their approximated MAP estimate *MAP propagation*. The pseudo-code of MAP propagation with gradient ascent to approximate $\hat{H}_t$ can be found in Algorithm 1 in Appendix A. Note that $N = 0$ recovers the special case of pure REINFORCE.

Similar to *actor-critic networks* [16], we can also train a critic network to estimate the state-value, $V^\pi(S_t)$, so $G_t$ in (9) can be replaced by TD error $\delta_t = R_t + \gamma V^\pi(S_{t+1}) - V^\pi(S_t)$ and the whole algorithm can be implemented online. To better facilitate temporal credit assignment, we can also use eligibility traces to replace the gradient in (9), using the same idea of actor-critic networks with eligibility trace [16]. The pseudo-code of it can be found in Algorithm 2 in Appendix A.

A team of agents can be trained by MAP propagation to estimate the state-value, such that a separate team of agents can fulfill the role of a critic network, and the whole actor-critic network can be trained without backprop. A simple way to achieve this is to convert the estimation of state-value to an RL task, but this conversion is inefficient since the information of optimal actions is lost (the agent only knows a scalar reward but not the target output). Appendix C proposes a new learning rule to train a team of agents to estimate the state-value by MAP propagation efficiently based on the information of optimal actions.

Essentially, MAP propagation is equivalent to applying REINFORCE after minimizing the energy function. As there are many studies on the biological plausibility of REINFORCE, we refer readers to the book of Sutton and Barto [16] for a review and discussion of the connection between REINFORCE and neuroscience. The main difference between MAP propagation and REINFORCE is the minimization of the energy function given by the update rule (8). This update rule is *local* as it only depends on the units one layer above and below based on feedforward and feedback connections. There is much evidence that feedback signals in brains alter neural activity [7, 18], supporting the use of feedback connections in MAP propagation. The update rule can also be performed in parallel for all layers, removing the need for precise coordination between feedforward and feedback connections as in backprop. However, the update rule requires the feedback weight to be symmetric of the feedforward weight and different values to be propagated through feedforward and feedback connections.

MAP propagation fits well into the recently proposed *NGRAD hypothesis* [7], which hypothesizes that the cortex uses differences in activity states to drive learning. The main idea of NGRAD is that "*higher-level activities can nudge lower-level activities towards values that are more consistent with the*

---

[1] We let $\hat{H}_t^0 = H_t^0 = S_t$, $\hat{H}_t^l$ denote the $l$th layer in $\hat{H}_t$ for $l \in \{1, 2, ..., L-1\}$, and $\hat{H}_t^L = H_t^L = A_t$.

*higher-level activity*", which also describes the process of energy minimization in MAP propagation. A detailed discussion of the biological plausibility of MAP propagation and its relationship with the NGRAD hypothesis can be found in Appendix F.

## 3.2 Relationship with Backpropagation

A network of stochastic units cannot be directly trained by backprop. However, assuming that there exists a re-parameterization of $H_t$ by $Z_t$ conditioned on $S_t$, denoted by $h(Z_t; W, S_t)$, then we can update parameters using backprop with the re-parameterization trick [19]; that is, for $l \in \{1, 2, ..., L\}$:

$$W^l \leftarrow W^l + \alpha G_t \nabla_{W^l} \log \pi_L(h^{L-1}(Z_t; W, S_t), A_t; W^L). \tag{10}$$

It can be shown that this learning rule follows the gradient of return in expectation (See Appendix B.5 for the proof). Using a similar argument as in MAP propagation, we can reduce the variance associated with the learning rule by minimizing the energy function before applying the learning rule.

Interestingly, for a network with all hidden layers being normally distributed, when the values of hidden layers are settled to a stationary point of the energy function, the parameter update given by backprop with the reparametrization trick in (10) is equivalent to the parameter update given by REINFORCE in (3):[2]

**Theorem 2.** *Let the policy be a multi-layer network of stochastic units with all hidden layers normally distributed as defined in Section 2. There exists a re-parameterization of $H_t$ by $Z_t$ conditioned on $S_t$ that is independent of $t$, denoted by $h(Z_t; W, S_t)$, such that for any $l \in \{1, 2, ..., L\}$, $s \in \mathcal{S}$, $\hat{h} \in \mathbb{R}^{n(1)} \times \mathbb{R}^{n(2)} \times ... \times \mathbb{R}^{n(L-1)}$, $\hat{z} \in \mathbb{R}^{n(1)} \times \mathbb{R}^{n(2)} \times ... \times \mathbb{R}^{n(L-1)}$ and $a \in \mathcal{A}$, if $\nabla_h E(\hat{h}; s, a) = 0$ and $\hat{z} = h^{-1}(\hat{h}; W, s)$, then*

$$\nabla_{W^l} \log \pi_l(\hat{h}^{l-1}, \hat{h}^l; W^l) = \nabla_{W^l} \log \pi_L(h^{L-1}(\hat{z}; W, s), a; W^L). \tag{11}$$

The proof is in Appendix B.2. In other words, by nudging the values of units in lower layers towards values that are more consistent with the value of units in the final layer, the parameter update given by REINFORCE becomes the same as backprop with the reparametrization trick. With $N \to \infty$ and $\alpha$ sufficiently small, under the update rule (7), $H_t$ will converge to the stationary point of the energy function. Therefore, the parameter update given by MAP propagation converges to the parameter update given by backprop with the reparametrization trick after minimizing the energy function.

Despite the close relationship between MAP propagation and backprop, there are key differences between the two algorithms. Compared to backprop, one major limitation of MAP propagation is that it can only be applied to RL tasks. MAP propagation is also more computationally expensive than backprop due to the minimization of the energy function in every step. However, MAP propagation can be applied to a network of discrete units. Moreover, MAP propagation does not require non-local feedback signals or precise coordination between feedforward and feedback pathways, which makes it more biologically plausible than backprop.

To see that MAP propagation is more computationally expensive than backprop, denote $L$ as the number of layers in the network and $N$ as the number of steps in the energy minimization (the inner loop). MAP propagation requires $LN$ layer updates during each step of energy minimization, while backprop requires $L$ layer updates to compute the feedback signals (iterating from the top layers). Therefore, MAP propagation takes $N$ times more layer updates than backprop. However, the $LN$ layer updates in MAP propagation can be done in parallel for all layers, so the time complexity for a single step of MAP propagation can be reduced to $\mathcal{O}(N)$ from $\mathcal{O}(LN)$ if the update is done in parallel. For backprop, parallel computation of feedback signals is not possible, so the time complexity for a single step of backprop remains $\mathcal{O}(L)$.

## 4 Related Work

Local learning rules based on MAP estimates of latent variables have been proposed in both unsupervised and supervised learning tasks. For unsupervised learning tasks, Bengio et al. [5] proposed training a deep generative model by using MAP estimate to infer the value of latent variables, conditioned on observed variables; in the same work, they also proposed to learn the feedback weights

---

[2]We let $\hat{h}^0 = s$, $\hat{h}^l$ denote the $l^{\text{th}}$ layer in $\hat{h}$ for $l \in \{1, 2, ..., L-1\}$, and $\hat{h}^L = a$.

such that the constraint of symmetric weight can be removed. This idea can also possibly be applied to our algorithm. For supervised learning tasks, Whittington and Bogacz [20] proposed training a deep network with local learning rules based on MAP inference and clipping the output value of the network to the target value. In contrast to these works, MAP propagation applies to RL tasks and does not require clipping any units' values.

Besides algorithms based on MAP estimates, many biologically plausible alternatives to backprop have been proposed. Biologically plausible learning rules based on reward prediction errors and attentional feedback have also been studied [21, 22, 23]; but these learning rules mostly require a non-local feedback signal. Moreover, there have been efforts to study local learning rules based on contrastive divergence or nudging the values of output units towards the target value [24, 25, 26]. See the work of Lillicrap et al. [7] for a comprehensive review of algorithms that approximate backprop with local learning rules based on the differences in units' values. Contrary to these works, MAP propagation requires neither the temporal difference in units' value nor multiple phases of learning.

Another perspective of training a multi-level network of stochastic units relates to viewing each unit as an RL agent, forming a hierarchy of agents. In hierarchical RL, Levy et al. [27] proposed learning a multi-level hierarchy with hindsight actions, which is similar to our idea of replacing the values of hidden layers with the MAP estimates. The special case of a team of agents forming a network to solve a task in a cooperative way was first proposed by Tsetlin [28] and later by Barto [29], and a comprehensive review can be found in chapter 15.10 of the book of Sutton and Barto [16]. Such a team of agents is recently called coagent networks [15] and theories relating to training coagent networks have been studied [15, 30, 31]. However, coagent networks learn much slower than an ANN trained by backprop due to the high variance associated with the learning rule. A few methods have been proposed to reduce this variance. Thomas [15] proposed to disable exploration of units randomly, but the learning speed is still not comparable to backprop. Chung [32] proposed the *Weight Maximization* algorithm, which replaces the global reward in the learning rule with a new local reward signal, and showed that this change of reward signal can reduce the variance effectively.

In addition, there is a large amount of literature on methods for training a network of stochastic units. A review can be found in the work of Weber et al. [33], which includes the re-parametrization trick [34] and REINFORCE [14]. They introduced methods to reduce the variance of the estimate, such as baseline and critic. These ideas are orthogonal to the use of MAP estimate to reduce the variance associated with REINFORCE.

## 5 Experiments

To test the algorithm, we first consider a single-time-step MDP that is similar to the multiplexer task [29]. This is to test the performance of the algorithm as an actor. Then we consider a scalar regression task to test the performance of the algorithm as a critic. Finally, we consider some standard RL tasks to test the performance of the algorithm as both an actor and a critic.

In the below tasks, all the teams of agents (or coagent networks) have the same architecture: a two-hidden-layer network, with the first hidden layer having 64 units, the second hidden layer having 32 units, and the output layer having one unit. All hidden layers are normally distributed with $\pi_l(H_t^{l-1}, \cdot; W^l) = N(f(W^l H_t^{l-1}), \sigma_l^2)$ for $l = 1, 2$, and $f(x) = \log(1 + \exp(x))$, the softplus function. For the network in multiplexer task and the actor network with discrete output, the output unit's distribution is given by the softmax function on the previous layer, i.e. $\pi_L(H^{L-1}, a; W^L) = \text{softmax}_a(W^L H^{L-1}/T)$, where $T > 0$ is a scalar hyperparameter representing the temperature. For the network in the scalar regression task, the critic network and the actor network with continuous output, the output unit's distribution is normally distributed with mean given by a linear transformation of the previous layer's value and a fixed variance, i.e. $\pi_L(H^{L-1}, \cdot; W^L) = N(W^L H^{L-1}, \sigma_L^2)$. We used $N = 20$ in MAP propagation. Other hyperparameters and details of experiments can be found in Appendix D.

### 5.1 Multiplexer Task

We consider a single-time-step MDP that is similar to the $k$-bit multiplexer task. In our single-time-step MDP, the state is sampled from all possible values of a binary vector of size $k + 2^k$ with equal probability. The action set is $\{-1, 1\}$, and we give a reward of 1 if the action of the agent is the

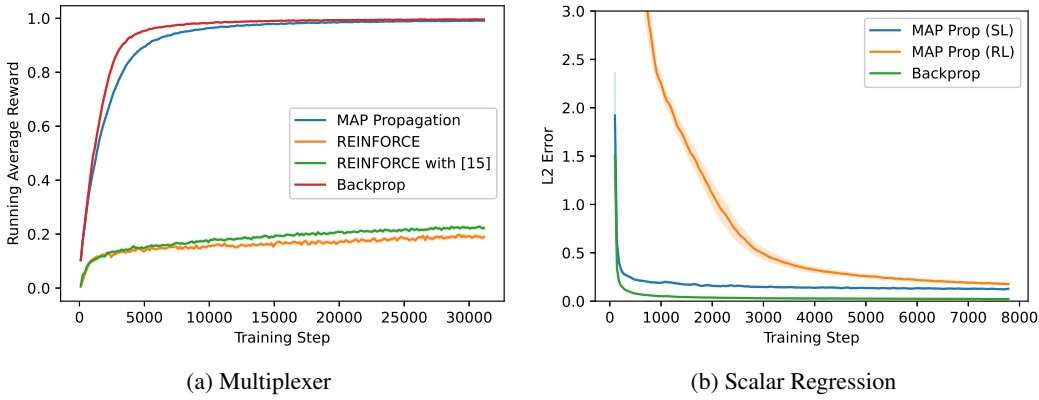

(a) Multiplexer

(b) Scalar Regression

Figure 1: Running average rewards over the last 10 episodes in multiplexer task and scalar regression task. Results are averaged over 10 independent runs, and shaded area represents standard deviation over the runs.

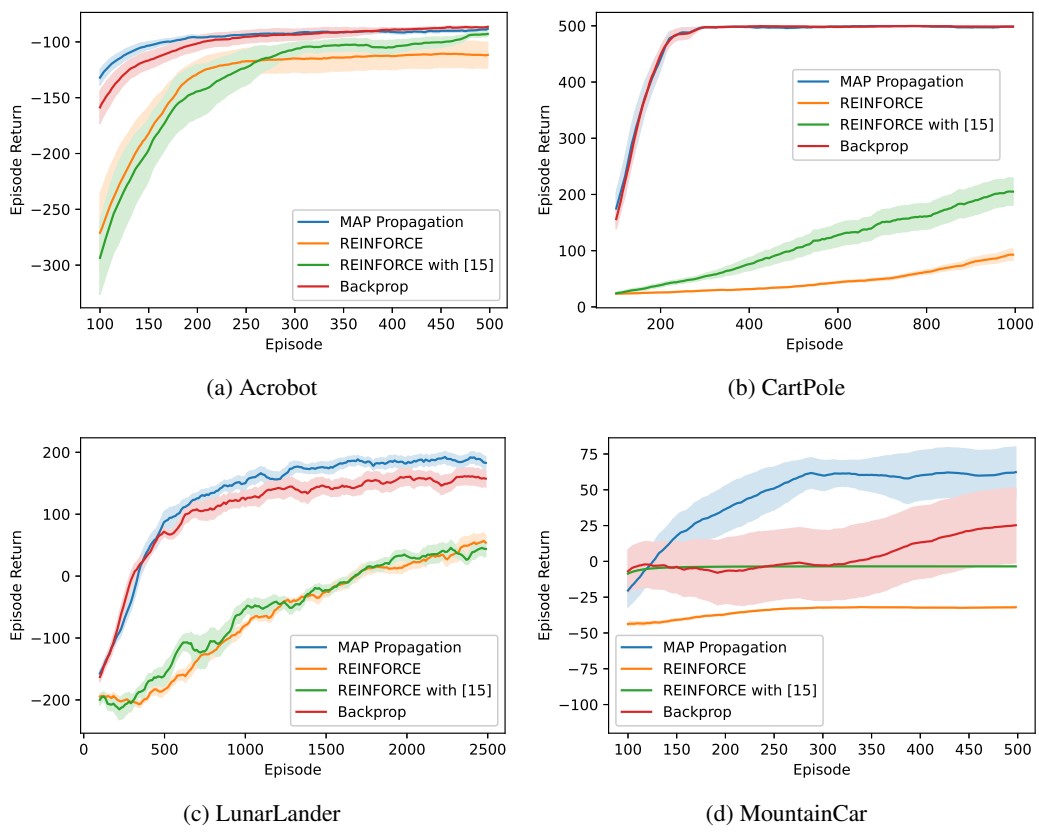

(a) Acrobot

(b) CartPole

(c) LunarLander

(d) MountainCar

Figure 2: Running average returns over the last 100 episodes in Acrobot, CartPole, LunarLander and MountainCar. Results are averaged over 10 independent runs, and shaded area represents standard deviation over the runs.

Table 1: Average return over all episodes.

| | Acrobot | | CartPole | | LunarLander | | MountainCar | |
|---|---|---|---|---|---|---|---|---|
| | Mean | Std. | Mean | Std. | Mean | Std. | Mean | Std. |
| MAP Propagation | **-100.29** | 5.40 | **459.70** | 13.89 | **127.88** | 24.57 | **39.45** | 30.48 |
| REINFORCE | -148.42 | 47.65 | 47.29 | 8.22 | -62.05 | 16.16 | -35.52 | 0.65 |
| REINFORCE with [15] | -149.11 | 33.50 | 112.58 | 42.54 | -54.61 | 23.47 | -4.65 | 0.21 |
| Backprop | -106.29 | 15.00 | 458.96 | 9.44 | 104.92 | 31.98 | 4.30 | 59.28 |

desired action and -1 otherwise. The desired action is given by the output of a multiplexer, with the input of the multiplexer being the state. We consider $k = 5$ here, so the dimension of the state space is 37.

We used Algorithm 1 for training a team of agents by MAP propagation. We consider three baselines: 1. REINFORCE - A team of agents trained entirely by REINFORCE; 2. REINFORCE with [15] - A team of agents trained entirely by REINFORCE but with the variance reduction method considered by Thomas [15]; 3. Backprop - An ANN with a similar architecture where the output unit is trained by REINFORCE and hidden units are trained by backprop.

The results are shown in Fig 1. We observe that MAP propagation performs much better than the two REINFORCE baselines. The result suggests that MAP propagation can improve the learning speed of REINFORCE significantly, such that its learning speed is comparable to backprop.

## 5.2 Scalar Regression Task

In the following, we consider a scalar regression task. The dimension of input is 8 and follows the standard normal distribution. The target output (a real scalar) is computed by a one-hidden-layer ANN with weights chosen randomly. The goal of the task is to predict the target output given the input.

For MAP propagation, we used Algorithm 1 and tested two variants. In the first variant that is labeled as 'MAP Prop (RL)', we treated the task as a single-time-step MDP with the negative L2 loss as the reward and trained the network using Algorithm 1. In the second variant that is labeled as 'MAP Prop (SL)', we replaced the learning rule in Algorithm 1 with the learning rule proposed in Appendix C, which incorporates the value of target output. For the baseline, we trained an ANN with a similar architecture by gradient descent on the L2 loss.

The results are shown in Fig 1. We observe that if we directly use the negative L2 loss as the reward, then the learning speed of MAP propagation is significantly lower than backprop since the information of target output is not incorporated. On the other hand, if we use the learning rule in Appendix C to incorporate the information of target output, then the learning speed of MAP propagation is comparable to backprop. However, the asymptotic performance of MAP propagation is slightly worse than backprop, which is due to the stochastic property of teams of agents. Nonetheless, this may not be a problem when applying MAP propagation to train a critic network, since the value function to be estimated is also constantly changing with the policy function.

## 5.3 Reinforcement Learning Task

In the following, we consider four standard RL tasks: Acrobot, CartPole, LunarLander, and continuous MountainCar in OpenAI's Gym. For MAP propagation, we use the actor-critic network with eligibility traces given by Algorithm 2, and the critic network is trained by the learning rule proposed in Appendix C. We consider three baselines. For the first and second baseline, the actor network is a team of agents trained entirely by REINFORCE. However, since REINFORCE cannot train a critic network directly and it is inefficient to convert the state-value estimation task to an RL task, we used an ANN with a similar architecture trained by backprop as the critic network. We also used the variance reduction method considered by Thomas [15] in the second baseline. We used eligibility traces in the training of both the actor and the critic network. For the third baseline, we used actor-critic with eligibility traces (episodic) [16] trained by backprop. Both actor and critic networks are ANNs with an architecture similar to the team of agents.

The average return over ten independent runs is shown in Fig 2. Let $\bar{G}$ denote the average return of all episodes. The mean and standard deviation of $\bar{G}$ over the ten runs can be found in Table 1. For all RL tasks, we observe that MAP propagation has a better performance than the baselines in terms of the average return $\bar{G}$. The result demonstrates that a team of agents trained with MAP propagation can learn much faster than a team of agents trained with REINFORCE, such that the team of agents can solve common RL tasks at a similar (or higher) speed compared to an ANN trained by backprop.

Although there are other algorithms besides actor-critic networks that can solve RL tasks more efficiently, the present work aims to compare different training methods for hidden units in an actor-critic network. Teams of agents trained by MAP propagation can also be applied to algorithms besides actor-critic networks, such as variants of actor-critic networks like Proximal Policy Optimization [35] and action-value based methods like Q-Learning [16].

We also notice that MAP Prop performs better than backprop on tasks where a high degree of exploration is required. For example, MAP propagation performs slightly worse than backprop on the multiplexer task but much better than backprop on the MountainCar task. This may suggest that teams of agents trained with MAP propagation can have better exploration than an ANN trained with backprop. This is further corroborated by the analysis of agents' behaviors on MountainCar, a task where an agent can easily be stuck in the local optima. For backprop, we found that agents in most of the runs are stuck in early episodes even with the use of entropy regularization [36]. In contrast, a team of agents trained by MAP propagation can reach the goal of the task successfully in all runs. A detailed analysis of this can be found in Appendix E. One possible explanation for the better exploration is that actions of agents in lower layers can be considered as abstract actions, and the exploration of these agents corresponds to exploration beyond the primitive actions.

## 6    Future Work and Conclusion

The ability to train teams of agents efficiently leads to many possible future directions. First, the local property of the learning rule points to the possibility of implementing MAP propagation asynchronously, such that it can be implemented efficiently with neuromorphic circuits [37]. Second, agents in the team can have a different temporal resolution, such that the actions of agents can be extended temporally and become options [38], yielding better exploration and learning behavior.

In conclusion, we propose a new algorithm that reduces the variance associated with REINFORCE and thus significantly increases the learning speed in the training of a team of agents. The proposed algorithm is also more biologically plausible than backprop while maintaining a comparable learning speed to backprop. Our work opens the prospect of the broader application of teams of agents in deep RL. Our experiments also suggest a team of agents trained by MAP propagation can have more sophisticated exploration compared to an ANN trained by backprop.

## 7    Acknowledgment

We would like to thank Andrew G. Barto, who inspired this research and provided valuable insights and comments, as well as Andy K.P. Chan for feedback and discussions.

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
