# A  Algorithms

**Algorithm 1:** MAP Propagation - Monte-Carlo Policy-Gradient Control

1   **Input:** differentiable policy function: $\pi_l(h^{l-1}, h^l; W^l)$ for $l \in \{1, 2, ..., L\}$;
2   **Algorithm Parameter:** step size $\alpha > 0$, $\alpha_h > 0$; update step $N \geq 1$;
3   discount rate $\gamma \in [0, 1]$;
4   **Initialize policy parameter:** $W^l$ for $l \in \{1, 2, ..., L\}$;
5   **Loop forever (for each episode):**
6      Generate an episode $S_0, H_0, A_0, R_1, ..., S_{T-1}, H_{T-1}, A_{T-1}, R_T$ following $\pi_l(\cdot, \cdot; W^l)$ for $l \in \{1, 2, ..., L\}$;
7      **Loop for each step of the episode,** $t = 0, 1, ..., T - 1$**:**
8          $G \leftarrow \sum_{k=t+1}^{T} \gamma^{k-t-1} R_k$ ;
         `/* MAP Gradient Ascent                                    */`
9          **for** $n := 1, 2, ..., N$ **do**
10              $H_t^l \leftarrow H_t^l + \alpha_h(\nabla_{H_t^l} \log \pi_l(H_t^{l-1}, H_t^l; W^l) + \nabla_{H_t^l} \log \pi_{l+1}(H_t^l, H_t^{l+1}; W^{l+1}))$ for $l \in \{1, 2, ..., L-1\}$;
11          **end**
         `/* Apply REINFORCE                                          */`
12          $W^l \leftarrow W^l + \alpha G \nabla_{W^l} \log \pi_l(H_t^{l-1}, H_t^l; W^l)$ for $l \in \{1, 2, ..., L\}$;

---

**Algorithm 2:** MAP Propagation - Actor Network with Eligibility Trace

1   **Input:** differentiable policy function: $\pi_l(h^{l-1}, h^l; W^l)$ for $l \in \{1, 2, ..., L\}$;
2   **Algorithm Parameter:** step size $\alpha > 0$, $\alpha_h > 0$; update step $N \geq 1$;
3   trace decay rate $\lambda \in [0, 1]$; discount rate $\gamma \in [0, 1]$;
4   **Initialize policy parameter:** $W^l$ for $l \in \{1, 2, ..., L\}$;
5   **Loop forever (for each episode):**
6      Initialize $S$ (first state of episode) ;
7      Initialize zero eligibility trace $\mathbf{z}^l$ for $l \in \{1, 2, ..., L\}$ ;
8      **Loop while $S$ is not terminal (for each time step):**
9          $H^0 \leftarrow S$ ;
         `/* 1.  Feedforward phase                                    */`
10          Sample $H^l$ from $\pi_l(H^{l-1}, \cdot; W^l)$ for $l \in \{1, 2, ..., L\}$ ;
11          $A \leftarrow H^L$;
         `/* 2.  REINFORCE phase                                      */`
12          **if episode not in first time step then**
13              Receive TD error $\delta$ from the critic network;
14              $W^l \leftarrow W^l + \alpha\delta\mathbf{z}^l$ for $l \in \{1, 2, ..., L\}$;
15          **end**
         `/* 3.  Minimize energy phase                                */`
16          **for** $n := 1, 2, ..., N$ **do**
17              $H^l \leftarrow H^l + \alpha_h(\nabla_{H^l} \log \pi_l(H^{l-1}, H^l; W^l) + \nabla_{H^l} \log \pi_{l+1}(H^l, H^{l+1}; W^{l+1}))$ for $l \in \{1, 2, ..., L-1\}$ ;
18          **end**
         `/* 4.  Trace accumluation phase                             */`
19          $\mathbf{z}^l \leftarrow \gamma\lambda\mathbf{z}^l + \nabla_{W^l} \log \pi_l(H^{l-1}, H^l; W^l)$ for $l \in \{1, 2, ..., L\}$;
20          Take action $A$, observe $S, R$ ;

---

# B  Proof

In the proofs below, we may omit the subscript $t$ whenever it is unnecessary. In addition to the notation in Section 2, we define $D_x f$ as the Jacobian matrix of $f$ w.r.t. $x$.

## B.1 Proof of Theorem 1

$$\mathbb{E}[G\nabla_{W^l}\log\Pr(A|S;W)] \tag{12}$$

$$=\mathbb{E}[\frac{G}{\Pr(A|S;W)}\nabla_{W^l}\Pr(A|S;W)] \tag{13}$$

$$=\mathbb{E}[\frac{G}{\Pr(A|S;W)}\nabla_{W^l}\sum_h \pi_L(h^{L-1},A;W^L)\pi_{L-1}(h^{L-2},h^{L-1};W^{L-1})...$$

$$\pi_2(h^1,h^2;W^2)\pi_1(S,h^1;W^1)] \tag{14}$$

$$=\mathbb{E}[\frac{G}{\Pr(A|S;W)}\sum_{h^l,h^{l-1}}\Pr(A,H^l=h^l,H^{l-1}=h^{l-1}|S)\nabla_{W^l}\log\pi_l(h^{l-1},h^l;W^l)] \tag{15}$$

$$=\mathbb{E}[G\sum_{h^{l-1},h^l}\Pr(H^{l-1}=h^{l-1},H^l=h^l|S,A)\nabla_{W^l}\log\pi_l(h^{l-1},h^l;W^l)] \tag{16}$$

$$=\mathbb{E}[G\,\mathbb{E}[\nabla_{W^l}\log\pi_l(H^{l-1},H^l;W^l)|S,A]] \tag{17}$$

$$=\mathbb{E}[\mathbb{E}[G|S,A]\,\mathbb{E}[\nabla_{W^l}\log\pi_l(H^{l-1},H^l;W^l)|S,A]] \tag{18}$$

$$=\mathbb{E}[\mathbb{E}[G\nabla_{W^l}\log\pi_l(H^{l-1},H^l;W^l)|S,A]] \tag{19}$$

$$=\mathbb{E}[G\nabla_{W^l}\log\pi_l(H^{l-1},H^l;W^l)]. \tag{20}$$

(17) to (18) uses the fact that, for any random variables $Z$ and $Y$, $\mathbb{E}[\mathbb{E}[Z|Y]f(Y)]=\mathbb{E}[Zf(Y)]$.
(18) to (19) uses the fact that $G$ is conditional independent of $H^l,H^{l-1}$ given $S,A$.
(19) to (20) uses the law of total expectation.

Note that (17) to (20) also shows the steps for (4).

## B.2 Proof of Theorem 2

Using $\nabla_h E(\hat{h};s,a)=0$, $\hat{h}^{L-1}$ can be expressed as:

$$\nabla_h E(\hat{h};s,a)=0, \tag{21}$$

$$-\nabla_{h^{L-1}}\log\pi(\hat{h}^{L-2},\hat{h}^{L-1};W^{L-1})=\nabla_{h^{L-1}}\log\pi(\hat{h}^{L-1},a;W^L), \tag{22}$$

$$\frac{1}{(\sigma^{L-1})^2}(\hat{h}^{L-1}-g^{L-1}(\hat{h}^{L-2};W^{L-1}))=\nabla_{h^{L-1}}\log\pi(\hat{h}^{L-1},a;W^L), \tag{23}$$

$$\hat{h}^{L-1}=g^{L-1}(\hat{h}^{L-2};W^{L-1})+(\sigma^{L-1})^2\nabla_{h^{L-1}}\log\pi(\hat{h}^{L-1},a;W^L). \tag{24}$$

And for $l=1,2,...,L-2$, we have:

$$\nabla_h E(\hat{h};s,a)=0, \tag{25}$$

$$-\nabla_{h^l}\log\pi(\hat{h}^{l-1},\hat{h}^l;W^l)=\nabla_{h^l}\log\pi(\hat{h}^l,\hat{h}^{l+1};W^{l+1}), \tag{26}$$

$$\frac{1}{(\sigma^l)^2}(\hat{h}^l-g^l(\hat{h}^{l-1};W^l))=\frac{1}{(\sigma^{l+1})^2}D_{h^l}g^{l+1}(\hat{h}^l;W^{l+1})^T(\hat{h}^{l+1}-g^{l+1}(\hat{h}^l;W^{l+1})), \tag{27}$$

$$\hat{h}^l=g^l(\hat{h}^{l-1};W^l)+\left(\frac{\sigma^l}{\sigma^{l+1}}\right)^2 D_{h^l}g^{l+1}(\hat{h}^l;W^{l+1})^T(\hat{h}^{l+1}-g^{l+1}(\hat{h}^l;W^{l+1})), \tag{28}$$

$$\hat{h}^l=g^l(\hat{h}^{l-1};W^l)+\left(\frac{\sigma^l}{\sigma^{l+2}}\right)^2 D_{h^l}g^{l+1}(\hat{h}^l;W^{l+1})^T D_{h^{l+1}}g^{l+2}(\hat{h}^{l+1};W^{l+2})^T$$

$$(\hat{h}^{l+2}-g^{l+2}(\hat{h}^{l+1};W^{l+2})), \tag{29}$$

$$\hat{h}^l=g^l(\hat{h}^{l-1};W^l)+(\sigma^l)^2 D_{h^l}g^{l+1}(\hat{h}^l;W^{l+1})^T D_{h^{l+1}}g^{l+2}(\hat{h}^{l+1};W^{l+2})^T$$

$$...D_{h^{L-2}}g^{L-1}(\hat{h}^{L-2};W^{L-1})^T\nabla_{h^{L-1}}\log\pi_L(\hat{h}^{L-1},a;W^L). \tag{30}$$

Substituting back to the REINFORCE update, we have:

$$\nabla_{W^l} \log \pi_l(\hat{h}^{l-1}, \hat{h}^l; W^l) \tag{31}$$

$$=\frac{1}{(\sigma^l)^2} D_{W^l} g^l(\hat{h}^{l-1}; W^l)^T(\hat{h}^l - g^l(\hat{h}^{l-1}; W^l)) \tag{32}$$

$$=D_{W^l} g^l(\hat{h}^{l-1}; W^l)^T D_{h^l} g^{l+1}(\hat{h}^l; W^{l+1})^T D_{h^{l+1}} g^{l+2}(\hat{h}^{l+1}; W^{l+2})^T ...$$
$$D_{h^{L-2}} g^{L-1}(\hat{h}^{L-2}; W^{L-1})^T \nabla_{h^{L-1}} \log \pi_L(\hat{h}^{L-1}, a; W^L). \tag{33}$$

We will show that the R.H.S. also equals (33). Consider the re-parameterization of $H^l$ by $Z^l$ conditioned on $H^{l-1}$ using $g^l(H^{l-1}; W^l) + \sigma^l Z^l$ and $Z^l$ are independent standard Gaussian noises for $l \in \{1, 2, ..., L-1\}$. Then by re-parameterizing all hidden layers, we can find $h(Z; W, S)$ that is the re-parameterization of $H$ by $Z := \{Z^1, Z^2, ..., Z^{L-1}\}$ conditioned on $S$. To be concrete, $h^l(Z; W, S) = g^l(h^{l-1}(Z; W, S); W^l) + \sigma^l Z^l$ for $l \in \{1, 2, ..., L-1\}$, and $h^0(Z; W, S) := S$.

Then, for $l \in \{1, 2, ..., L-2\}$, we have:

$$\nabla_{W^l} \log \pi_L(h^{L-1}(z; W, s), a; W^L) \tag{34}$$

$$=D_{W^l}(h^{L-1}(z; W, s))^T \nabla_{h^{L-1}} \log \pi_L(h^{L-1}(z; W, s), a; W^L) \tag{35}$$

$$=D_{W^l}(g^{L-1}(h^{L-2}(z; W, s); W^{L-1}) + \sigma^{L-1} z^{L-1})^T \nabla_{h^{L-1}} \log \pi_L(h^{L-1}(z; W, s), A; W^L) \tag{36}$$

$$=D_{W^l}(h^{L-2}(z; W, s))^T D_{h^{L-2}} g^{L-1}(h^{L-2}(z; W, s); W^{L-1})^T$$
$$\nabla_{h^{L-1}} \log \pi_L(h^{L-1}(z; W, s), a; W^L) \tag{37}$$

$$=D_{W^l} g^l(h^{l-1}(z; W, s); W^l)^T D_{h^l} g^{l+1}(h^l(z; W, s); W^{l+1})^T D_{h^{l+1}} g^{l+2}(h^{l+1}(z; W, s); W^{l+2})^T ...$$
$$D_{h^{L-2}} g^{L-1}(h^{L-2}(z; W, s); W^{L-1})^T \nabla_{h^{L-1}} \log \pi_L(h^{L-1}(z; W, s), a; W^L). \tag{38}$$

If we evaluate (34) at $z = \hat{z}$, then (38) becomes (33), which completes the proof for $l \in \{1, 2, ..., L-1\}$. The proof for $l = L$ is similar and is omitted here.

### B.3   Proof of Theorem 3

Similar to the proof of Theorem 2, for $l \in \{1, 2, ..., L-1\}$, the L.H.S. can be expressed as:

$$\frac{A^*(s) - \hat{\mu}^L}{a - \hat{\mu}^L} \nabla_{W^l} \log \pi_l(\hat{h}^{l-1}, \hat{h}^l; W^l) \tag{39}$$

$$=\frac{A^*(s) - \hat{\mu}^L}{a - \hat{\mu}^L} D_{W^l} g^l(\hat{h}^{l-1}; W^l)^T D_{h^l} g^{l+1}(\hat{h}^l; W^{l+1})^T D_{h^{l+1}} g^{l+2}(\hat{h}^{l+1}; W^{l+2})^T ...$$
$$D_{h^{L-2}} g^{L-1}(\hat{h}^{L-2}; W^{L-1})^T \nabla_{h^{L-1}} \log \pi_L(\hat{h}^{L-1}, a; W^L) \tag{40}$$

$$=\frac{A^*(s) - \hat{\mu}^L}{(\sigma^L)^2} D_{W^l} g^l(\hat{h}^{l-1}; W^l)^T D_{h^l} g^{l+1}(\hat{h}^l; W^{l+1})^T D_{h^{l+1}} g^{l+2}(\hat{h}^{l+1}; W^{l+2})^T ...$$
$$D_{h^{L-2}} g^{L-1}(\hat{h}^{L-2}; W^{L-1})^T \nabla_{h^{L-1}} g^L(\hat{h}^{L-1}; W^L). \tag{41}$$

We then prove that the R.H.S. also equals (41). Consider the same re-parameterization of $H$ by $Z := \{Z^1, Z^2, ..., Z^{L-1}\}$ conditioned on $S$, denoted by $h(Z; W, S)$, as in the proof of Theorem 2. Then, for $l \in \{1, 2, ..., L-1\}$,

$$\nabla_{W^l} - (A^*(s) - g^L(h^{L-1}(z; W, s); W^L))^2 \tag{42}$$

$$=2(A^*(s) - g^L(h^{L-1}(z; W, s); W^L)) \nabla_{W^l} g^L(h^{L-1}(z; W, s); W^L) \tag{43}$$

$$=2(A^*(s) - g^L(h^{L-1}(z; W, S); W^L)) D_{W^l}(h^{L-1}(z; W, s))^T \nabla_{h^{L-1}} g^L(h^{L-1}(z; W, s); W^L) \tag{44}$$

$$=2(A^*(s) - g^L(h^{L-1}(z; W, S); W^L)) D_{W^l} g^l(h^{l-1}(z; W, s); W^l)^T D_{h^l} g^{l+1}(h^l(z; W, s); W^{l+1})^T$$
$$D_{h^{l+1}} g^{l+2}(h^{l+1}(z; W, s); W^{l+2})^T ... \nabla_{h^{L-1}} g^L(h^{L-1}(z; W, s); W^L). \tag{45}$$

If we evaluate (42) at $z = \hat{z}$, then (45) becomes proportional to (41) with a ratio $2(\sigma^L)^2$, which completes the proof for $l \in \{1, 2, ..., L-1\}$. The proof for $l = L$ is similar and is omitted here.

## B.4 Details of (7) to (8)

$$\nabla_{H_t^l} \log \Pr(H_t | S_t, A_t) \tag{46}$$

$$= \nabla_{H_t^l} \left( \log \Pr(H_t, A_t | S_t) - \log \Pr(A_t | S_t) \right) \tag{47}$$

$$= \nabla_{H_t^l} \log \Pr(H_t, A_t | S_t) \tag{48}$$

$$= \nabla_{H_t^l} \log \left( \prod_{i=0}^{L-1} \pi_i(H_t^i, H_t^{i+1}; W^{i+1}) \right) \tag{49}$$

$$= \nabla_{H_t^l} \log \pi_{l+1}(H_t^l, H_t^{l+1}; W^{l+1}) + \nabla_{H_t^l} \log \pi_l(H_t^{l-1}, H_t^l; W^l). \tag{50}$$

## B.5 (10) Follows the Gradient of Return in Expectation

We will show that the learning rule given by (10) follows the gradient of return in expectation. For $l \in \{1, 2, ..., L-1\}$:

$$\nabla_{W^l} \mathbb{E}[G] \tag{51}$$

$$= \mathbb{E}[G \nabla_{W^l} \log \Pr(A | S; W)] \tag{52}$$

$$= \mathbb{E}[\frac{G}{\Pr(A | S; W)} \nabla_{W^l} \Pr(A | S; W)] \tag{53}$$

$$= \mathbb{E}[\frac{G}{\Pr(A | S; W)} \nabla_{W^l} \sum_z \Pr(A | Z = z, S; W) \Pr(Z = z | S)] \tag{54}$$

$$= \mathbb{E}[\frac{G}{\Pr(A | S; W)} \sum_z \Pr(A | Z = z, S; W) \Pr(Z = z | S) \nabla_{W^l} \log \Pr(A | Z = z, S; W)] \tag{55}$$

$$= \mathbb{E}[G \sum_z \Pr(Z = z | S, A; W) \nabla_{W^l} \log \Pr(A | Z = z, S; W)] \tag{56}$$

$$= \mathbb{E}[G \, \mathbb{E}[\nabla_{W^l} \log \Pr(A | Z, S; W) | S, A]] \tag{57}$$

$$= \mathbb{E}[G \nabla_{W^l} \log \Pr(A | Z, S; W)] \tag{58}$$

$$= \mathbb{E}[G \nabla_{W^l} \log \pi_L(h^{L-1}(Z; W, S), A; W^L)]. \tag{59}$$

(51) to (52) uses REINFORCE and other steps are similar to those in the proof of Theorem 1.

## B.6 Variance Reduction of (4)

We will show that for $l \in \{1, 2, ..., L\}$:

$$\mathrm{Var}[G \, \mathbb{E}[\nabla_{W^l} \log \pi_l(H^{l-1}, H^l; W^l) | S, A]] \leq \mathrm{Var}[G \nabla_{W^l} \log \pi_l(H^{l-1}, H^l; W^l)]. \tag{60}$$

The proof is as follows:

$$\mathrm{Var}[G \nabla_{W^l} \log \pi_l(H^{l-1}, H^l; W^l)] \tag{61}$$

$$= \mathrm{Var}[\mathbb{E}[G \nabla_{W^l} \log \pi_l(H^{l-1}, H^l; W^l) | S, A, G]]$$
$$+ \mathbb{E}[\mathrm{Var}[G \nabla_{W^l} \log \pi_l(H^{l-1}, H^l; W^l) | S, A, G]] \tag{62}$$

$$\geq \mathrm{Var}[\mathbb{E}[G \nabla_{W^l} \log \pi_l(H^{l-1}, H^l; W^l) | S, A, G]] \tag{63}$$

$$= \mathrm{Var}[G \, \mathbb{E}[\nabla_{W^l} \log \pi_l(H^{l-1}, H^l; W^l) | S, A, G]] \tag{64}$$

$$= \mathrm{Var}[G \, \mathbb{E}[\nabla_{W^l} \log \pi_l(H^{l-1}, H^l; W^l) | S, A]]. \tag{65}$$

(61) to (62) uses the law of total variance.
(62) to (63) uses the fact that the second term must be non-negative.
(64) to (65) uses the fact that $G$ is conditional independent with $H$ given $S$ and $A$.

## C  MAP Propagation for Critic Networks

We consider how to apply MAP propagation to a critic network. As the function of a critic network can be seen as approximating the scalar value $R_t + \gamma \hat{v}(S_{t+1})$, we first consider how to learn a scalar regression task by MAP propagation in general.

A scalar regression task can be converted into a single-time-step MDP with the appropriate reward and $\mathbb{R}$ as the action set. For example, we can set the reward function to be $R(S, A) = -(A - A^*(S))^2$ (we drop the subscript $t$ as it only has a single time step), where $A \in \mathbb{R}$ denotes the output of the network and $A^*(S) \in \mathbb{R}$ denotes the target output given input $S$. The maximization of rewards in this MDP is equivalent to the minimization of the L2 distance between the predicted value and the target value.

But this conversion is inefficient since the information of the reward function is lost. In the following discussion, we restrict our attention to a network of stochastic units where all hidden layers and the output layer are normally distributed as defined in Section 2. Let $\mu^L$ be the conditional mean of the output layer; that is, $\mu^L = g(H^{L-1}; W^L)$. For this network, we propose an alternative learning rule that is similar to REINFORCE but with the return $G$ replaced by $(A^*(S) - \mu^L)/(A - \mu^L)$; that is, for $l \in \{1, 2, ..., L\}$:

$$W^l \leftarrow W^l + \alpha \frac{A^*(S) - \mu^L}{A - \mu^L} \nabla_{W^l} \log \pi_l(H^{l-1}, H^l; W^l). \tag{66}$$

It can be shown that after minimizing the energy function, the learning rule (66) for the network is equivalent to gradient descent on the L2 error by backprop with the re-parameterization trick:

**Theorem 3.** *Let the policy be a multi-layer network of stochastic units with all hidden layers normally distributed as defined in Section 2 and the output layer has a single unit. There exists a re-parameterization of $H$ by $Z$ conditioned on $S$, denoted by $h(Z; W, S)$, such that for any $A^* : \mathcal{S} \to \mathbb{R}$, $l \in \{1, 2, ..., L\}$, $s \in \mathcal{S}$, $\hat{h}, \hat{z} \in \mathbb{R}^{n(1)} \times \mathbb{R}^{n(2)} \times ... \times \mathbb{R}^{n(L-1)}$ and $a \in \mathbb{R}$, if $\nabla_h E(\hat{h}; s, a) = 0$ and $\hat{z} = h^{-1}(\hat{h}; W, s)$, then*

$$\frac{A^*(s) - \hat{\mu}^L}{a - \hat{\mu}^L} \nabla_{W^l} \log \pi_l(\hat{h}^{l-1}, \hat{h}^l; W^l) \propto -\nabla_{W^l} \left(A^*(s) - \tilde{\mu}^L\right)^2, \tag{67}$$

*where $\hat{\mu}^L := g^L(\hat{h}^{L-1}; W^L)$ and $\tilde{\mu}^L := g^L(h^{L-1}(\hat{z}; W, s); W^L)$.*

Therefore, we apply the learning rule (66) after minimizing the energy function by (8). The pseudo-code is the same as Algorithm 1 in Appendix A, but with $G$ replaced by $(A^*(S) - \mu^L)/(A - \mu^L)$ in line 12.

We apply the above method to train a critic network, where the output of the network, $A_t \in \mathbb{R}$, is an estimation of the current value. In a critic network, the target output is $R_t + \gamma A_{t+1}$. However, a more stable estimate of target output is $R_t + \gamma \mu_{t+1}^L$ since the difference between $A_{t+1}$ and $\mu_{t+1}^L$ is an independent Gaussian noise that can be removed. Therefore, we define the target output, denoted by $A^*(S)$, to be $R_t + \gamma \mu_{t+1}^L$ and the TD error, denoted by $\delta_t$, to be $R_t + \gamma \mu_{t+1}^L - \mu_t^L$. Substituting back into (66), the learning rule for the critic network becomes:

$$W^l \leftarrow W^l + \alpha \frac{\delta_t}{A_t - \mu_t^L} \nabla_{W^l} \log \pi_l(H_t^{l-1}, H_t^l; W^l), \tag{68}$$

which is almost the same as the update rule for the actor network except the additional denominator $A_t - \mu_t^L$. The pseudo-code of training a critic network with eligibility trace using MAP propagation is the same as Algorithm 2 in Appendix A, except (i) line 13 is replaced with $\delta \leftarrow \gamma \mu + R - \mu'$ where $\mu = g^L(H^{L-1}; W^L)$ and $\mu'$ is $\mu$ in the previous time step, and (ii) the gradient term in line 19 is multiplied by $(A - \mu)^{-1}$.

Both the critic and the actor network can be trained together, and the TD error $\delta$ computed by the critic network can be passed to the actor network in line 13 of Algorithm 2.

## D  Details of Experiments

In the multiplexer task, there are $k + 2^k$ binary inputs, where the first $k$ bits represent the address and the last $2^k$ bits represent the data, each of which is associated with an address. The output of

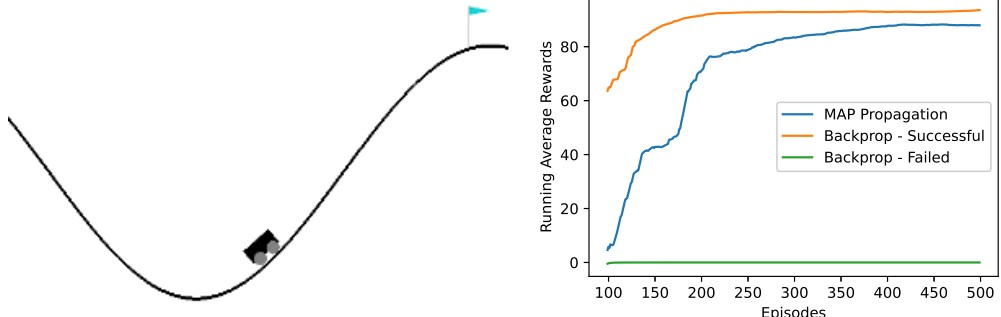

Figure 3: Illustration of MountainCar.

Figure 4: Running average rewards over the last 100 episodes of the selected runs in MountainCar.

the multiplexer is given by the value of data associated with the address. This is similar to the 2-bit multiplexer considered by Barto [1].

We used Algorithm 1 in the multiplexer and the scalar regression experiment, and the hyperparameters can be found in Table 2. We used a different learning rate $\alpha$ for each layer of the network, and we denote the learning rate for the $l^{\text{th}}$ layer as $\alpha_l$. We denote the variance of the Gaussian distribution for the $l^{\text{th}}$ layer as $\sigma_l^2$. The step size of hidden units when minimizing energy, $\alpha_h$, is selected to be 0.5 times the variance of the unit. For the learning rule in line 12 of the pseudo-code, we used Adam optimizer [2] instead, with $\beta_1 = 0.9$ and $\beta_2 = 0.999$. We used batch update in both tasks, which means that we compute the parameter update for each sample in a batch, then we update the parameter using the average of these parameter updates. These hyperparameters are selected based on manual tuning to optimize the learning curve. We did the same manual tuning for the baseline models.

We used Algorithm 2 to train both the critic and the actor network in the experiments on RL tasks, and the hyperparameters can be found in Table 3. We did not use any batch update in our experiments, and we used a discount rate of $0.98$ for all tasks. The step size of hidden units when minimizing energy, $\alpha_h$, is selected to be 0.5 times the variance of the unit. We used Acrobot-v1, CartPole-v1, LunarLander-v2, and MountainCarContinuous-v0 in OpenAI's Gym for the implementation of the RL tasks.

For the update rules in line 14 of Algorithm 2, we used Adam optimizer instead, with $\beta_1 = 0.9$ and $\beta_2 = 0.999$. Again, these hyperparameters are selected based on manual tuning to maximize the average return across all episodes. We did the same manual tuning for the baseline models.

For the ANNs in the baseline models, the architecture is similar to the team of agents: 64 units on the first hidden layer and 32 units on the second hidden layer. If the output range is continuous, the output layer is a linear layer. If the output range is discrete, the output layer is a softmax layer. We used the softplus function as the activation function in the ANNs, which performed similarly to the ReLu function in our experiments.

We annealed the learning rate linearly such that the learning rate is reduced to $\frac{1}{10}$ of the initial learning rate at $50000$ and $100000$ steps in CartPole and Acrobot respectively, and the learning rate remains unchanged afterward. We also annealed the learning rate linearly for the baseline models. We found that this can make the final performance more stable. For LunarLander and MountainCar, we did not anneal the learning rate. For MountainCar, we bounded the reward by $\pm 5$ to stabilize learning.

## E    Experiments on MountainCar

In the continuous version of MountainCar, the state is composed of two scalar values, which are the position and the velocity of the car, and the action is a scalar value corresponding to the force applied to the car. The goal is to reach the peak of the mountain on the right, where a large positive reward is given. However, to reach the peak, the agent has to first push the car to the left in order to gain enough momentum to overcome the slope. A small negative reward that is proportional to the force

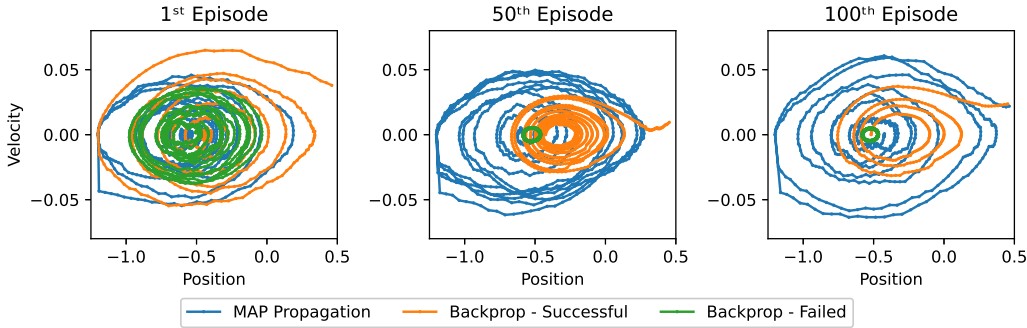

Figure 5: State trajectories of the 1st, 50th and 100th episode of the selected runs. If the position is larger than 0.45, then the agent reaches the goal. Although the team of agents trained by MAP propagation did not reach the goal in both the 1st and 50th episode, the team was still exploring a large portion of the state space, which is in contrast to the failed baseline that stayed at the center. For the baseline, if it does not reach the goal in the first several episodes, it will be stuck in the local optima.

applied is given on every time step to encourage the agent to reach the goal with minimum force. An illustration of MountainCar is shown in Fig 3.

One locally optimal policy is to apply zero force at every time step so that the car always stays at the bottom. In this way, the return is zero since no force is applied. We found that in many runs, the ANN trained by backprop was stuck in this locally optimal policy. However, in a few runs, the ANN can accidentally reach the goal in early episodes, which makes the ANN able to learn quickly and achieve an asymptotic reward of over +90, slightly higher than that of a team of agents trained by MAP propagation. The learning curve of a successful and a failed run of the agent trained by backprop is shown in Fig 4.

In contrast, for teams of agents trained by MAP propagation, the teams in all runs can learn a policy that reaches the goal successfully. However, this comes at the expense of slower learning and a slightly worse asymptotic performance, as seen from the learning curve of a typical run of the team of agents trained by MAP propagation shown in Fig 4. This is likely due to the larger degree of exploration in MAP propagation, which can be illustrated by the state trajectories shown in Fig 5.

We used the same variance for the output unit in both MAP propagation and the baseline models. We found that even using a larger variance or adding entropy regularization cannot prevent the baseline models from being stuck in the local optima. This suggests that MAP propagation allows more sophisticated exploration instead of merely more exploration. In a team of agents, each agent in the team is exploring its own action space, thus allowing exploration in different levels of the hierarchy. This may explain the difference in the exploration behavior observed in a team of agents trained by MAP propagation compared to an ANN trained by backprop.

## F  Biological Plausibility of MAP Propagation

As discussed in the paper, backprop has three major problems with biological plausibility due to the requirement of 1. non-local information in the learning rule, 2. precise coordination between the feedforward and feedback phase, and 3. symmetry of feedforward and feedback weights. However, REINFORCE does not have any of these issues. Other than the global reinforcement signal, the learning rule of REINFORCE does not depend on non-local information. Also, since REINFORCE does not require any feedback connections, the second and third issues do not exist for REINFORCE. We refer readers to chapter 15 of the book of Sutton and Barto [3] for a review and discussion of the connection between REINFORCE and neuroscience.

Compared to backprop, REINFORCE is more consistent with biologically-observed forms of synaptic plasticity. When applied to a Bernoulli-logistic unit, REINFORCE gives a three-factor learning rule which depends on a reinforcement signal, input, and output of the unit. This is similar to R-STDP observed biologically, which depends on a neuromodulatory input, presynaptic spikes,

and postsynaptic spikes. It has been proposed that dopamine, a neurotransmitter, plays the role of neuromodulatory input in R-STDP and corresponds to the TD error from RL.

Despite the elegance of REINFORCE, its learning speed is far lower than backprop and scales poorly with the number of units in the network since only a scalar feedback is used to assign credit to all units in the network. It can be argued that learning speed may not be the issue for the biological plausibility of REINFORCE, given that evolution already equips the brain with prior knowledge, and the brain can learn from experience accumulated over the entire lifetime. However, the learning speed of REINFORCE may not explain many remarkable learning behaviors of humans, such as mastering Go, despite the fact that the ability to play Go likely does not come from prior knowledge shaped by evolution. Given billions of neurons in the brain, it is likely that the brain employs some forms of structural credit assignment to speed up learning.

MAP propagation presents one possible solution for structural credit assignment. Essentially, MAP propagation is equivalent to applying REINFORCE after minimizing the energy function. The idea of minimizing the energy function is to nudge the values of hidden units towards those that are more consistent with the values of units on the first and last layer, i.e. the state and the action. In this process, feedback connections are required to drive the value of units, so as to propagate information from the layers above.

Although the purpose of feedback connections in the brain is still not completely understood, there has been evidence showing that feedback connections can drive the activity of neurons [4, 5, 6]. For example, in the visual system, the activity of neurons that is responsible for the selected action will be enhanced by feedback connection [4]. This is analogous to the updates of hidden units in MAP propagation. The general idea that feedback connection drives the activity of units in lower layers to facilitate local learning rules underlies many proposals for biological learning and machine learning algorithms [7]. This idea is also fundamental to the NGRAD hypothesis [7], which will be discussed next.

NGRAD hypothesis is a recently proposed hypothesis that unifies many biologically plausible alternatives to backprop with local learning rules [8, 9, 10, 11, 12, 13, 14, 15, 16]. It hypothesizes that the cortex uses the differences in activity states to drive learning, and the induced differences are brought by the nudging of lower-level activities towards those values that are more consistent with the high-level activities. In this way, local learning rules can yield an approximation to backprop without storing units' values and error signals at the same time.

MAP propagation fits well into the NGRAD hypothesis. When normally distributed hidden units are settled to the minima of the energy function, REINFORCE, a local learning rule except for the global reinforcement signal, yields the same parameter update given by backprop. Therefore, MAP propagation can be seen as an approximation of backprop by changing the values of hidden units. However, MAP propagation has major differences with many other algorithms based on the NGRAD hypothesis (NGRAD algorithms). First, most NGRAD algorithms require storage of past units' values (e.g. in target propagation [16], the unit has to store its past value to compute the reconstruction error) or separate phases of learning (e.g. the positive and negative phase in contrastive divergence [11]), but MAP propagation requires neither of them. Second, most NGRAD algorithms require precise coordination between feedforward and feedback connections (e.g. in target propagation, the unit has to coordinate between computing the reconstruction error and adding it to the uncorrelated target). In contrast, the updates for all layers can be done in parallel without any coordination between feedforward and feedback connections in MAP propagation. Third, MAP propagation is derived based on RL, while NGRAD algorithms are derived based on either supervised or unsupervised learning. Given the observation of R-STDP in biological systems and the correspondence between R-STDP and REINFORCE, MAP propagation presents a new paradigm of explaining biological learning in NGRAD algorithms. However, a major limitation of MAP propagation is that it requires a different value to be propagated through feedforward and backward connections.

To see this, we will closely examine the update rule (8) for minimizing energy function in MAP propagation. Assuming all units are normally distributed with a fixed variance; i.e. $\pi_l(H^{l-1}, \cdot; W^l) = N(f(W^l H^{l-1}), \sigma^2)$ for $l \in \{1, 2, ..., L\}$ and $f$ is a non-linear activation function, then the update

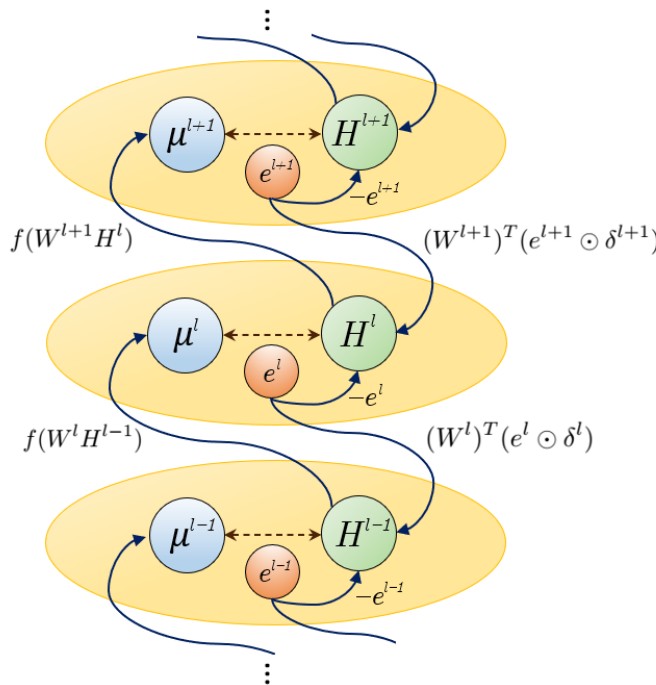

Figure 6: Illustration of minimizing the energy function in MAP propagation. The mean value of units on the layer $l$, denoted by $\mu^l$, is computed as a function of the value of units on the previous layer, denoted by $H^{l-1}$. The difference between the current and the mean value of units, denoted by $e^l$, is then used to drive the value of units on the same layer ($H^l$) and the value of units on the previous layer ($H^{l-1}$).

rule (8) and the learning rule (3) becomes [1]:

$$\Delta H^l = \frac{1}{\sigma^2} \left( -e^l + (W^{l+1})^T (e^{l+1} \odot \delta^{l+1}) \right), \tag{69}$$

$$\Delta W^l = \frac{1}{\sigma^2} \left( G \cdot e^l (H^{L-1})^T \right), \tag{70}$$

where $\mu^l = f(W^l H^{l-1})$, $\delta^l = f'(W^l H^{l-1})$, $e^l = H^l - \mu^l$ for $l \in \{1, 2, ..., L\}$.

Both the update rule (8) and the learning rule (3) are local and can be applied to all hidden layers in parallel. There are two components in the update rule: i. the feedforward signal $-e^l$ and ii. the feedback signal $(W^{l+1})^T (e^{l+1} \odot \delta^{l+1})$. The feedforward signal nudges the value of the unit, $H^l$, closer to the mean value of the unit, $\mu^l$, which only depends on feedforward signals. However, it is not yet clear how the feedback signal can be implemented with biological systems. First, it requires information to be propagated through the feedback weight $(W^{l+1})^T$ that is symmetric of the feedforward weight in the next layer, which may not be biologically plausible. Nonetheless, recent work has shown that symmetric weights may not be necessary for backprop due to the phenomenon of 'feedback alignment' [17, 18, 19], and similar phenomenons may also exist for MAP propagation. Second, the information to be propagated backward is $e^{l+1} \odot \delta^{l+1}$, which is different from the information to be propagated forward ($H^l$). An illustration of this is shown in Fig 6.

The issue of propagating two different values also exists for backprop since error signals, instead of units' values, are propagated backward in backprop. However, in backprop, the backpropagated error signals have to be stored separately from the units' values, so as to compute the next error signals to be passed to the lower layers. In contrast, the feedback signal in MAP propagation is only used to nudge the value of units and does not need to be stored separately. In other words, backprop requires non-local information in the computation of feedback signal, but not MAP propagation.

---

[1]We ignore the subscript $t$ here since it does not affect our discussion, and $\odot$ denotes element-wise multiplication.

It is not yet clear how a neuron can propagate two different values at the same time, even if both values are locally available. But there are many possible solutions to avoid propagating different values in MAP propagation. For instance, since the feedback signal $(W^{l+1})^T(e^{l+1} \odot \delta^{l+1})$ can be expressed as a function of $H^l$ and $H^{l+1}$, it might be possible to approximate this term based on feedback connections (for $H^{l+1}$) and recurrent connections (for $H^l$), and learn the weights in these connections, such that all units are propagating the same value. Another possible solution is to minimize the energy function by hill-climbing methods instead of gradient ascent, such that only the scalar energy, instead of feedback connection, is required to guide the minimization of the energy function. Further work can be done on these possible solutions.

Despite the limitations of MAP propagation, we argue that MAP propagation is more biologically plausible than backprop. The two major limitations of MAP propagation also exist in backprop, but MAP propagation does not require non-local information in both the learning rule or the computation of feedback signals. Also, the update of all layers can be done in parallel in MAP propagation, removing the requirement of precise coordination between feedforward and feedback connections that is required in backprop.

Table 2: Hyperparameters used in multiplexer and scalar regression experiments.

|  | Multiplexer | Scalar Regression |
|---|---|---|
| Batch Size | 128 | 128 |
| $N$ | 20 | 20 |
| $\alpha_1$ | 4e-2 | 6e-2 |
| $\alpha_2$ | 4e-5 | 6e-5 |
| $\alpha_3$ | 4e-6 | 6e-6 |
| $\sigma_1^2$ | 0.3 | 0.0075 |
| $\sigma_2^2$ | 1 | 0.025 |
| $\sigma_3^2$ | n.a. | 0.025 |
| $T$ | 1 | n.a. |

Table 3: Hyperparameters used in Acrobot, Cartpole, Lunarlander and MountainCar experiments.

|  | Acrobot | | CartPole | | LunarLander | | MountainCar | |
|---|---|---|---|---|---|---|---|---|
|  | Critic | Actor | Critic | Actor | Critic | Actor | Critic | Actor |
| $N$ | 20 | 20 | 20 | 20 | 20 | 20 | 20 | 20 |
| $\alpha_1$ | 2e-2 | 1e-2 | 2e-2 | 1e-2 | 1e-2 | 4e-3 | 1e-2 | 4e-3 |
| $\alpha_2$ | 2e-5 | 1e-5 | 2e-5 | 1e-5 | 1e-5 | 4e-6 | 1e-5 | 4e-6 |
| $\alpha_3$ | 2e-6 | 1e-6 | 2e-6 | 1e-6 | 1e-6 | 4e-7 | 1e-6 | 4e-7 |
| $\sigma_1^2$ | 0.06 | 0.03 | 0.03 | 0.03 | 0.003 | 0.06 | 0.003 | 0.03 |
| $\sigma_2^2$ | 0.2 | 0.1 | 0.1 | 0.1 | 0.01 | 0.2 | 0.01 | 0.1 |
| $\sigma_3^2$ | 0.2 | n.a. | 0.1 | n.a. | 0.01 | n.a. | 0.05 | 0.5 |
| $T$ | n.a. | 4 | n.a. | 2 | n.a. | 8 | n.a. | n.a. |
| $\lambda$ | .97 | .97 | .95 | .95 | .97 | .97 | .97 | .97 |