# OpenReview forum: "MAP Propagation Algorithm: Faster Learning with a Team of Reinforcement Learning Agents"
_NeurIPS.cc/2021/Conference — NeurIPS 2021 Poster_

### Official Review · Reviewer_NswB · 2021-07-02

**Rating:** 7
**Confidence:** 2

**Summary:**

The paper introduces a new synapse-local method for reinforcement learning, that is similar to the REINFORCE algorithm, but reduces the high variance caused by REINFORCE's need to assess many stochastic samples, at the cost of added computation.

IIUC, it does this by 1- making REINFORCE layer-local (Eq 2), 2- Replacing the gradient of probabilities of observed responses  by its expectation given the state and action (which is uncomputable) (Eq 4-5), and 3- Approximating this uncomputable expectation by replacing actual activations with their maximum a posteriori estimate, given the chosen action and observed state. The latter turns out to be computable, with some iterative process.

The algorithm is also shown to be equivalent, update-wise, to plain backpropagation with the reparametrization trick (this is how I understood Theorem 2 and Eq. 11, but I may be wrong, see below).

Various experiments show that this method learns faster than REINFORCE and about as fast as backprop with the reparametrization trick, in terms of number of steps (though each step is computationally more intensive).

**Limitations And Societal Impact:**

Limitations of the algorithm, especially its higher computational cost, seem adequately stated (e.g. in last paragraph of section 3.2).

**Main Review:**

I found the paper interesting, though somewhat confusing to me, possibly due to my lack of expertise with the details of the algorithms.

To my knowledge, the method is novel, and addresses an important limitation of REINFORCE-like algorithms (the high variance and the difficulties this causes for learning). The *relative* biological plausibility (the rule is local to each neuron, though the actual mechanism seems somewhat involved for neurons to perform) is a nice bonus.

The experiments are well chosen and the results are informative.

The paper is well-written throughout, though could gain some clarity.

First and foremost, it would be *extremely helpful* to provide a very short, intuitive summary of the overall proposal and derivation (similar to the two first paragraphs in the Summary above, but of course more accurate) before going on with the full description, so that the reader would get a general idea of where this is going and why we're doing it before trudging through the whole derivation in section 3.1.

The introduction mentions what MAP does (lines 41-42 of p. 2) , but not *why*. IIUC, it is so that we need fewer actual samples to estimate the gradients. In fact, the crucial justification for the whole approach seems to that the RHS in Eq. 4 should have lower variance than Eq. 5. It's not quite obvious why, and lines 116-117 in p.3 don't clarify much. The reason for the reduction of variance should be explained a bit better, possibly in an appendix section.

Also, when the introduction and abstract mention that the proposed approach is equivalent to backpropagation. IIUC, it should be "backpropagation with the reparametrization trick". If I am correct, this should be explicitly stated! (And if not, it should be clarified what exactly "backpropagation" means here).

A few other things that confused me:

* Eq 11 and Theorem 2 seem to assert that the method is equivalent to backprop with the reparametrization trick. But the experiments seem to show that the method has quite distinct properties, e.g. being better when more exploration is needed. Evidently I am missing something. Could this be clarified?

* For the regression part (p. 4, l. 146), I do not understand what information is "lost". The appendix seems to suggest a simple centering and normalization method to address this "loss". Again, this could be clarified.



**Time Spent Reviewing:**

4

---

> ### Author Response · Authors · 2021-08-09
> **Response to R4**
>
> We thank the reviewer for the constructive comments and detailed feedback. Below are our responses to the comments.
>
> R4: It would be extremely helpful to provide a very short, intuitive summary of the overall proposal and derivation
>
> We will add a brief overview of the algorithm in section 3 before going into the mathematical details.  We agree that it should help readers get an overall picture of the algorithm and improve the readability of the paper.
>
> R4: The introduction mentions what MAP does (lines 41-42 of p. 2) , but not why. IIUC, it is so that we need fewer actual samples to estimate the gradients.
>
> As correctly pointed out by the reviewer, the reason for using MAP estimate is to lower the variance of the update as justified by (4). The reason why it decreases variance can be explained by the law of total variance. Consider we are interested in estimating $ \mathbb{E} [f(X, Y)]$, where $X$ and $Y$ are random variables. Both estimators $f(X, Y)$ and $\mathbb{E} [f(X, Y)|Y]$ are unbiased estimators. By the law of total variance:
> $$\text{Var}(f(X, Y)) = \text{Var}(\mathbb{E}[f(X, Y)|Y]) + \mathbb{E}(\text{Var} [f(X, Y)|Y]) \geq \text{Var} (\mathbb{E}[f(X, Y)|Y]).$$
> Thus, the estimator $\mathbb{E} [f(X, Y)|Y]$ has a lower variance than the estimator $f(X, Y)$. In our case, $X$ corresponds to $H$ (the hidden state), and $Y$ corresponds to $S, A$ (the state and action). This explains why the terms inside the outer expectation in R.H.S. of (4) has a lower variance than that of L.H.S. We will add this discussion in the appendix.
>
> R4: Also, when the introduction and abstract mention that the proposed approach is equivalent to backpropagation. IIUC, it should be "backpropagation with the reparametrization trick".
>
> Since backprop with reparameterization trick on a network of normally distributed hidden units is equivalent to backprop on a network of deterministic hidden units with independently and normally distributed noise added to hidden units, we did not explicitly differentiate the two algorithms in the introduction. We acknowledge that it is unclear to readers without this discussion, so we will clarify the equivalence to be “backprop with the reparameterization trick”.
>
> R4: Eq 11 and Theorem 2 seem to assert that the method is equivalent to backprop with the reparametrization trick.
>
> For a network with normally distributed hidden units, MAP propagation is equivalent to backprop with the reparameterization trick after minimizing the energy function. That is, if we run (8) repeatedly until all units’ values converge, then at this set of units’ values, REINFORCE will give the same update as backprop with the reparameterization trick. Thus, if we directly apply backprop with the reparameterization trick without running (8) before, the update will not be the same as MAP propagation (note that MAP propagation is REINFORCE after running (8) repeatedly).
>
> R4: For the regression part (p. 4, l. 146), I do not understand what information is "lost". The appendix seems to suggest a simple centering and normalization method to address this "loss". Again, this could be clarified.
>
> When converting a regression task to an RL task, the negative loss is used as the reward to the agent, and usually, it is the negative L2 distance between the target output and the network output. Consider the case that the agent's output is 5.5 and the target output is 6.5. The reward to the agent is thus -1.0. However, the agent does not know whether it should increase or decrease the output (if the target output is 4.5, then the reward is still -1.0). However, in the original regression task, the optimal output (6.5) will be known to the network, and it can learn to increase its output. Therefore, in the conversion, the optimal information is lost, which makes learning slower.

---

### Official Review · Reviewer_pdvp · 2021-07-16

**Rating:** 7
**Confidence:** 4

**Summary:**

This paper proposes a local learning rule as a biologically plausible alternative to backprop. The core idea is to train each neuron using REINFORCE and mitigate the high variance issue of REINFORCE using their novel MAP propagation algorithm. They also show how a similar method could be used to train a value function for an actor-critic algorithm. And they are able to demonstrate the effectiveness of their method on few tasks.

**Limitations And Societal Impact:**

Yes.

**Main Review:**

Introducing a biologically plausible alternative to backprop is a very active research area that would be beneficial in bridging the gap between neuroscience and artificial neural networks. And often the proposed alternatives are sub-optimal and not able to match the performance of backprop. The results presented in this work show that their proposed method is able to reliably work. And the algorithm is relatively easy to implement and does not impose many constraints on the network architecture, hence could be a work that the community can build upon it.

In the first paragraph of related work, is the main novelty of this work compared to the cited works is the application of MAP propagation to RL as opposed to (un)supervised learning? Would be great to elaborate more.

While the text is well-written and I was able to follow the text, the main text is quite dense. Perhaps some restructuring and rewriting could make the paper more easily readable.

Please have a brief description of the multiplexer and scalar regression tasks to make the text self-sufficient.

Due to inner loop optimization for each layer, there is added computational complexity. Could you please also have a comparison of the computational complexity of this method vs backprop?

The appending L:82 mentions that a different learning rate was used for each layer. This will result in an increase in the number of hyper-parameters with layer size. How difficult was it to tune each of these separately?


**Time Spent Reviewing:**

3

---

> ### Author Response · Authors · 2021-08-09
> **Response to R3**
>
> We thank the reviewer for the constructive comments and detailed feedback. Below are our responses to the comments.
>
> R3: is the main novelty of this work compared to the cited works is the application of MAP propagation to RL as opposed to (un)supervised learning?
>
> Yes, one of the main novelties of MAP propagation compared to [5] and [20] is its application on RL tasks instead of supervised learning or unsupervised learning tasks. MAP propagation does not assume that the correct output is known and only requires a scalar reward that evaluates the network’s output. As a result, the learning rule of MAP propagation is substantially different to [5] and [20] despite the common use of MAP estimation in part of the algorithms. For example, MAP propagation does not require clamping any values of the units that is required in both [5] and [20].
>
> R3: Perhaps some restructuring and rewriting could make the paper more easily readable.
>
> We will add a brief overview of the algorithm in section 3 before going into the mathematical details, as suggested by R4. This should help readers get an overall picture of the algorithm.
>
> R3: Please have a brief description of the multiplexer and scalar regression tasks to make the text self-sufficient.
>
> A detailed description of the multiplexer task can be found in Appendix D. We will elaborate more about the scalar regression task in the revised version of the paper.
>
> R3: Due to inner loop optimization for each layer, there is added computational complexity. Could you please also have a comparison of the computational complexity of this method vs backprop?
>
> We will include the following discussion in the revised version of the paper: Denote $L$ as the number of layers in the network and $N$ as the number of steps in the energy minimization (the inner loop). MAP propagation requires $LN$ layer’s updates during the energy minimization, while backprop requires $L$ layer’s updates to compute the feedback signals (iterating from the top layers). Therefore, MAP propagation takes $N$ times more layer’s updates than backprop. However, the $LN$ layer’s updates in MAP propagation can be done in parallel for all layers, so the time complexity for a single step of MAP propagation is $\mathcal{O}(N)$ instead of $\mathcal{O}(LN)$ if the update is done in parallel. For backprop, parallel computation of feedback signals is not possible, so the time complexity for a single step of backprop is $\mathcal{O}(L)$.
>
> R3: The appending L:82 mentions that a different learning rate was used for each layer. This will result in an increase in the number of hyper-parameters with layer size. How difficult was it to tune each of these separately?
>
> It is not difficult to tune the different learning rates for each layer. We found that it is better for deeper layers to have a lower learning rate, and the optimal relationship between learning rates of different layers is very similar across tasks. For example, the learning rate of a layer divided by that of the next layer is the same in all our experiments, as shown in Table 3. So, we only have to tune a single scalar that multiples the learning rates of all layers in our experiments.

---

### Official Review · Reviewer_SPS7 · 2021-07-17

**Rating:** 7
**Confidence:** 3

**Summary:**

This work introduces a local learning rule (with a global reward signal). The basic idea is just a neural network with stochastic units and REINFORCE update each layer (that is, the REINFORCE gradient estimate is not backpropogated through the network, each layer computes an update weighted by the reward. This, by itself, is (as recognized in the paper) an old idea.

They key addition here is an approach to reducing the (prohibitive) variance of this approach by conditioning the update of a particular layer on the state and action (eq 5). This lower-variance update is then approximated by eq 6. One interpretation of this is that the activations of the layer being updated are shifted to be more compatible (lower free energy) with the nearby layers.

They try this approach on some standard simple RL benchmark tasks and find that it performs comparably to backprop but much better then naive REINFORCE (per layer) or a previous method for reducing variance in this situation.

**Limitations And Societal Impact:**

I don't think this paper raises particular concerns.

**Main Review:**

Originality:
As the author's recognize, the idea of apply REINFORCE to every unit in a network is old. As far as I know the variance reduction approach used here is novel and the main contribution.

Quality:

The approach appears technically sound (I did not check the appendix carefully) and the experimental results reasonable, although only on relatively simple tasks.

The approach introduce here is interesting, because any new learning rule is interesting. But its main claimed benefits are that is more biologically plausible (than backprop) due to the use of only local information and a global reward. For this reason, it seems that other somewhat biological plausible approaches should have also been included as baselines, such as feedback alignment [11-13].

This submission would also be strengthened by discussing how ``biologically plausible'' this learning method is. For example, (approximately) taking the argmax (eq 8) is done layer-wise rather than unit-wise. Is there a plausible neural mechanisms for this? (I don't know the answer).

Clarity:

Overall the paper is well written.

One minor thing is that in some cases I think it would be helpful to be explicit about what the expectation is taken over.

Significance:

Learning rules that are competitive (in learning speed) with backprop while using only local information are of interest to the community. Based on the experiment here, this approach appears of interest. The interest to the neuroscience community would depend on how biologically plausible this learning rule is when examined in more detail.

Minor Issues:

$H$ is referred to the as the ``value'' of a layer. It might be clearer to call this the activation of a layer, as ``value'' could mean weight or activation.

I realize that MAP is a common abbreviation, but I still think that it would be ideal to write out the term in full once.

**Time Spent Reviewing:**

3

---

> ### Author Response · Authors · 2021-08-09
> **Response to R2**
>
> We thank the reviewer for the constructive comments and detailed feedback. Below are our responses to the comments.
>
> R2: it seems that other somewhat biological plausible approaches should have also been included as baselines
>
> As the aim of our experiments is to demonstrate the effectiveness of MAP propagation in reducing the variance of REINFORCE, we did not include other biologically plausible learning rules as baselines in the experiments. Also, since most biologically plausible learning rules proposed have a sub-optimal performance compared to backprop and the main goal of these approaches is usually biological plausibility instead of performance [11-13, 21-26], it may not be very insightful to compare the performance between them directly. This may also explain why most experiments in these papers are compared against backprop only.
>
> R2: this submission would also be strengthened by discussing how ``biologically plausible'' this learning method is
>
> We refer the reviewer to Appendix F for a detailed discussion of the biological plausibility of MAP propagation, which discusses the possible neural mechanisms implementing equation (8). Though (8) is presented layer-wise, communication between units within the same layer is not required, as seen by (63). In other words, (8) and (63) can be performed unit-wise instead of layer-wise.
>
> R2: it would be helpful to be explicit about what the expectation is taken over
>
> We agree that some equations with multiple expectations (like (4)) may be confusing to readers. We will explicitly state what the expectation is taken over in the revised version.
>
> Other response:
>
> We will clarify H to be the activation value of a layer. Maximum a posteriori (MAP) is stated in L121, but we will repeat the full term in the introduction.

---

> > ### Comment · Reviewer_SPS7 · 2021-08-13
> > **Response to author's comments**
> >
> > I think that author's for their response and pointing me to appendix F which indeed is helpful.
> >
> > I'm not sure I entirely agree with the argument against including other "biologically plausible" baselines.
> >
> > Overall, I was already positive about the paper so I'm leaving my scores unchanged.

---

### Official Review · Reviewer_LSFF · 2021-07-17

**Rating:** 5
**Confidence:** 1

**Summary:**

The authors in the paper propose a method to train multi-layer neural networks.
The standard back-propagation algorithm aims at computing the influence of the output
neuron in a step wise chain rule fashion. The alternative proposed in this paper
is to compute the influence of a mid-layer neuron on the output of the network
using REINFORCE algorithm.  The proposal is to treat each neuron as a separate
learning agent. The authors main contribution is to implement MAP propagation to
control variance that comes with REINFORCE algorithm. The initial set of results show
that this technique can be comparable to back-propagation in RL settings.

**Ethical Concerns:**

None in my opinion

**Limitations And Societal Impact:**

The limitations have been addressed. I don't see any direct societal impacts in this paper.


**Main Review:**

I don't see any obvious technical faults in the paper.

My question to the authors is around the larger goal of this paper.
I don't see why being close to biological learning,  is so important if the
only goal is to achieve lower training error.

**Time Spent Reviewing:**

5 hrs

---

> ### Author Response · Authors · 2021-08-09
> **Response to R1**
>
> We thank the reviewer for the comments. The reviewer questions the motivation of proposing biologically plausible learning methods instead of learning methods that give a better empirical performance. This is a general question that involves the whole research field of biologically plausible machine learning (ML). Since the motivation of biologically plausible ML is a huge topic, we will be brief in the following discussion and refer the reviewer to [1] for a comprehensive review of the recent advances in biologically plausible ML and their significance in ML.
>
> First, although biologically plausible learning methods may not achieve state-of-the-art empirical performance in ML tasks, they often become the foundation for new state-of-the-art algorithms. An early example is Hopfield Networks [2], which were inspired by biological neural networks. Despite having a low memory capacity, Hopfield Networks led to Boltzmann Machines, Deep Belief Networks, and subsequently deep learning. Similarity, MAP propagation, as a biologically plausible alternative to backprop, may also lead to new state-of-the-art algorithms given the importance of backprop in deep learning. Second, biologically plausible learning methods often lead to advances in neuroscience and numerous examples are discussed in [1]. As pointed out by R3, MAP propagation may be of interest to the neuroscience community since it presents an alternative to backprop, which is generally regarded to be biologically implausible.
>
> Overall, although it is sound to question the motivation of biologically plausible learning rules in general, we do not agree that this is a sufficient reason to reject a paper.
>
> [1] Hassabis, Demis, et al. "Neuroscience-inspired artificial intelligence." Neuron 95.2 (2017): 245-258.
>
> [2] Hopfield, John J. "Neural networks and physical systems with emergent collective computational abilities." Proceedings of the national academy of sciences 79.8 (1982): 2554-2558.

---

### Decision · Program_Chairs · 2021-09-27

**Decision:**

Accept (Poster)

**Comment:**

This paper considers the problem of biologically plausible learning rules. For this, it looks at a local learning rule that uses REINFORCE to learn local updates for individual neurons. The main contribution is a novel variance reduction scheme for REINFORCE. While there were some fundamental question raised on biologically plausible learning rules (which is an active area of research), the reviewers found this a significant contribution and the paper interesting. As such, I would recommend acceptance of this paper.